# Stress and Psychological Well-Being in Military Gendarmes

**Maria Nicoleta Turliuc \*** and **Ana-Diana Balcan**

Department of Psychology, Alexandru Ioan Cuza University, 700554 Iasi, Romania;
ana.racoveanu@student.uaic.ro
* Correspondence: turliuc@uaic.ro

**Abstract:** Most studies consider the stressors faced by military personnel during operations in war zones and less those in peacetime activities. Work-related stress is a significant determinant of psychological well-being, but more relevant are the nature of stressors that military personnel is facing and the factors than can explain the relationship between work stress and well-being. The purpose of the present study was twofold: to examine the longitudinal relationships between organizational stress (OrgS), operational stress (OpS), and psychological well-being (PWB), and the mediating role of social support and coping mechanisms in the peacetime activities of police military gendarmes. A convenience sample of 210 military gendarmes (96.1% men and 3.90% women) completed five self-report scales regarding OrgS, OpS, PWB, social support, and coping mechanisms. All the variables were measured twice, in December 2021 (T1) and four months later in April 2022 (T2). The mean age was 38.52 years and the mean duration of military service was 14.52 years. The results show that baseline perceived stress, organizational and operational, has a significantly negative effect on PWB after four months. Perceived social support (at both T1 and T2) has a significant mediating role in the relationship between OrgS and PWB, as well as in the relationship between OpS and PWB. Among the coping mechanisms, only self-control (at T2) acts as a significant mediator of the relationship between OrgS and PWB. These findings could contribute to the development of intervention programs to increase the PWB of this personnel category, by working not only on perceived OrgS and OpS, but also on perceived social support and coping mechanisms.

**Keywords:** organizational stress; operational stress; psychological well-being; mediators; military gendarmes

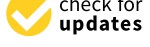



## 1. Introduction

Professions that involve managing other people's concerns, such as healthcare as well as law enforcement, generate particularly high levels of stress compared to other professions (Finn and Tomz 1998). The work of military personnel, including military gendarmes, is generally considered risky, involving exposure to critical situations, aversive events, and highly dangerous missions that have a negative impact on their mental health (Nielsen et al. 2020; Stogner et al. 2020). The Gendarmerie is the military branch of the police, with law enforcement duties among the civilians. In addition to their law enforcement duties, gendarmes can also undertake military defence functions in the event of war, and they have often been deployed in external inter-state conflicts (Lutterbeck 2004). Gendarmes interact with people belonging to different social groups, from people with mental disorders to criminals, and are frequently exposed to aversive situations, from verbal to physical aggression, and to potentially traumatic situations, ranging from assault to murder (He et al. 2002). They are constantly interacting with citizens, exposed to their issues, and exposed to physical or verbal aggression. In addition to operational stress (OpS) factors, gendarmes deal with organizational stress (OrgS) factors, such as the militarized structure, bureaucracy, lack of support from their coworkers or bosses, difficulties with promotion, a plethora of disciplinary procedures and restrictions, and an excessive work volume or administrative attributes (Pflanz 2001). Moreover, studies have indicated an association between higher

work-related stress and higher mental disorders among military staff members (e.g., Pflanz and Sonnek 2002; Pflanz 2001). OpS and OrgS can have a significant effect on the well-being of military employees (Ortega et al. 2007), on their physical and mental health.

The link between mission-induced stress and the well-being of military personnel in war zones and peace-keeping missions has been very well documented. However, there are very few studies concerning the stress experienced by military personnel during peacetime (Pflanz 2001), and no analyses regarding the stress in peacetime and its impact on well-being exist in the case of gendarmes from southeastern European counties. Additionally, the explanatory mechanisms of this relationship are not yet sufficiently investigated. Moreover, most of the previous research investigated the link between work-related stress and well-being cross-sectionally, and less longitudinally. This topic is extremely relevant, with military personnel of the Gendarmerie representing the main force that ensures the defense of life, bodily integrity, and freedom of the person, performing missions to ensure and restore public order, and anti-terrorist intervention missions. The Gendarmerie has a unique status in Romanian society, being both an institution that ensures security and one with a symbolic capital, a unique form of cultural militarism. In the context of the end of the millennium, a civil–military gap begins to manifest in military organizations (Ruckert 2006), a dissociation from the military ethos in favor of a new individualistic, post-modern ethos. The military is beginning to become more and more similar to civilian organizations (Lebel et al. 2022). In the context of these transformations, the study analyzes the perspective of the sense of commitment in professional relationships, the meaning of the profession, the context that military gendarmes feel as a discourse of chemistry that offers reward, in the form of belonging to the supportive relationships that define the fighting spirit.

To our knowledge, this is the first study to investigate the longitudinal relationship between OrgS and OpS and the psychological well-being (PWB) of gendarmes in a southeastern European country during their peacetime activity. The strengths of the present study also lie in the use of a longitudinal design, and in the identification of some of the explanatory mechanisms of these relationships, such as perceived social support, seeking social support, positive reappraisal, and self-control. More specifically, the negative longitudinal correlation between baseline work-related stress, OrgS and OpS, experienced by military gendarmes and their PWB four months later was examined. Furthermore, it was investigated whether social support and coping mechanisms, measured at baseline and four months apart, act as mediators in these relationships. The findings could contribute to the development of intervention programs to increase the PWB of military gendarmes, by working not only on perceived OrgS and OpS, but also on perceived social support and on different coping mechanisms.

*Stress and Psychological Well-Being in Military Personnel of the Gendarmerie, during Peacetime*

Psychological well-being represents individuals' satisfaction with their own life, their mental health, happiness, hope, and positive perception of their own self (Ryff 1989). This depends on individual experiences, and its aspects refer to personal affect, ability, and self-perception (Campbell et al. 1996). Psychological well-being is extremely important for the success of military operations, for the mental and physical health of military personnel (Chen et al. 2018). Stress generates a decrease in PWB (Quick and Henderson 2016) and affects the quality of experience and the perception of life (Harter et al. 2003). Stress is caused by individuals' effort to adapt to a dynamic environment when demanding occurrences are perceived as threatening or as surpassing their personal resources (Turliuc and Măirean 2014).

The specific requirements of the job may be broadly similar for peacetime police and military gendarmerie personnel, even though there may be significant differences in the nature of training, the intensity of operational demands, or the manner of intervention. Within the classical challenge–hindrance framework (Cavanaugh et al. 2000), work demands can be simultaneously hindering or challenging. Beyond individual characteristics and resources, the response to stressors also depends on the balance between challenges

and obstacles, organizational conditions, or burden limits (Gerich and Weber 2020). The literature has identified two major types of work-related stress encountered by police and military personnel: organizational and operational stress (Storch and Panzarella 1996). OrgS refers to problematic organizational aspects, such as the lack of trust in leaders, difficulties in communication, or organizational changes. These represent situations in which work-related stress interacts with individuals' physical and mental well-being and can disrupt their normal behavior, such as working in shifts, insufficient personnel, lack of appreciation, large volume of documents, negative public image (Violanti et al. 2016), bureaucracy, lack of perceived support from leaders, lack of promotion opportunities, management style, and lack of administrative support (Toch 2002). Brooks et al. (2017) identified the following sources of OrgS in military personnel: the lack or unhealthy state of support from leaders, lack of group cohesion, harassment and discrimination, role conflicts, excessive commitments, imbalance between effort and reward, specific requests, and imbalance between personal/family life and professional life.

OpS refers to stress factors that are specific to the activity undertaken, such as exposure to danger, physical threat, or unexpected events (Stephens and Long 2000). The aspects that can generate OpS are continuous exposure to citizens' dissatisfaction, use of force, violence, encountering dangerous and unexpected situations and even death threats (He et al. 2002). Law enforcement also implies facing poverty, suffering, physical threat, or death (Tadje 2014). Although there is evidence that exposure to stress contributes to the development of mental (Chou et al. 2014) and physical (Magnavita et al. 2018) health problems, research does not provide a conclusive picture of the risk to which military gendarmes are exposed.

In certain situations, the stress associated with the workplace directly influences well-being (i.e., the direct effect model, Wheaton 1985). Garbarino et al. (2013) mentioned that working in stressful conditions leads to the dissatisfaction and exhaustion of police officers, affecting their mental well-being and professional efficiency. Similarly, Ortega et al. (2007) underlined that police officers work in a unique environment, being exposed to traumatic events that have negative effects on their emotional and physical well-being. Given the negative health consequences of stress for both police and other law enforcement professions (Steptoe and Kivimaki 2012; Olsen et al. 2013), the following hypothesis was proposed:

**Hypothesis 1.** *Baseline work-related stress, OrgS and OpS, significantly and negatively predict PWB four months apart.*

The mediating role of social support

Social support represents the degree to which an individual perceives that he/she can count on one or more close people's help and one of the most used external resources in times of need. Perceived social support stems from the individual's perception of the availability of social relationships and is different from the support received (Dunkel-Schetter and Bennett 1990). According to Cohen (2004), social support represents a social network that provides psychological and material resources meant to help the individual manage stressful situations.

Stress researchers have underlined the significant role of social support (Cohen and Wills 1985; Dignam and West 1988; Hobfoll et al. 1990; Viswesvaran et al. 1999). Stress can have a main effect on social support, and vice versa (e.g., when people are too stressed, they are not able to identify all the available social support versus when receiving social support, stressors are either not perceived or are felt with a reduced intensity). Additionally, social support was identified to be directly associated with improved well-being (Cohen and Wills 1985; Garbarino et al. 2013). Moreover, the literature has identified two types of impacts of social support on the link between stress and its consequences. Social support can have a buffering effect so that the negative effects of stress are less felt (e.g., the buffering model, Cohen and Wills 1985). Individuals who have developed supportive social relationships generally feel better in comparison to those who do not (Westman 1992), even when they

are confronted with stress. Additionally, social support can have a significant mediating role in the link between stress and its outcomes. The negative effect of stress on social support and mental well-being is explained in the *model of damaged social support* (Dean and Ensel 1982). This indicates that negative life experiences can damage one's perception of the available social support and diminish one's mental health. Although numerous studies have demonstrated the negative relationship between stress and social support (Mitchell and Moos 1984; Dignam et al. 1986), there are not enough recent data to clarify the nature and intensity of stressors that can cause a deterioration of social support. Many forms of stressors, such as workplace tensions, are uncontrollable events (McFarlane et al. 1983) and could generate a deterioration in the perception of social support. Social support has pervasive benefits throughout adulthood (Shin and Park 2022). However, its importance could be lost over time due to changes in age or social circumstances associated (Carstensen 2006) with the military profession. It is difficult to assess the importance of different psychological needs for psychological well-being (Vermote et al. 2022), and at the level of the military gendarmerie population this information is missing. Using PWB as an outcome, stress will be negatively associated with perceived social support and well-being, while social support will have a positive association with PWB. Based on the reviewed literature and on the damaged social support model (Dean and Ensel 1982), we assumed that:

**Hypothesis 2.1.** *Baseline social support mediates the relationship between organizational or operational stress and PWB.*

**Hypothesis 2.2.** *Four months apart, social support still mediates the relationship between organizational or operational stress and PWB.*

The mediating role of coping mechanisms

Coping represents an individual's cognitive and behavioral effort to manage both external and internal requirements that surpass individual resources to overcome stressful situations (Folkman and Lazarus 1984). According to Lazarus and Folkman's (1984) *transactional model of stress*, people respond to stress with either problem-focused coping or emotion-focused coping. The response may include coping mechanisms based on emotions, which aim to reduce the feeling of tension, and strategies based on problem solving, which aim towards reducing the magnitude of the stressor itself (Carver and Connor-Smith 2010). Moreover, a recent meta-analysis (Hagger et al. 2017) has shown that adaptive strategies, including more proactive strategies focused on issues, predict positive results regarding mental and physical health, meaning that they reduce the progression of disorders, reduce stress, and increase well-being.

Police personnel use a variety of coping mechanisms, some adaptive and some less adaptive, to overcome stress (Hart et al. 1994). Active behaviors include discussions with coworkers, receiving counseling, and physical exercises. Less adaptive behaviors include the abuse of alcohol, withdrawal from friendships and relationships with family, and suicide (Richmond et al. 1998). If the number of studies investigating the mediating role of coping mechanisms in the relationship between perceived stress and well-being is relatively large, there are fewer studies related to military professions. Ryu et al. (2020) showed not only that job stress is associated with coping mechanisms and subjective well-being but also that some coping styles (e.g., problem solving and assistance pursuit) were mediating variables between stress and subjective well-being.

Research has shown that different coping strategies are associated with different ways of psychological adaptation (Finstad et al. 2021). First, emotion-focused coping, such as positive refocusing, can facilitate meaning making and posttraumatic growth (Tuncay and Musabak 2015). Second, self-control was associated with a higher level of cognitive well-being and positive affect (Hofmann et al. 2014). Third, in contrast to perceived social support, discussed previously, the search for social support—as a coping mechanism—represents the tendency to seek social support when needed and research

suggests that people who actively seek support may be more effective in managing traumatic stressors (Chao 2011). Empirical research also indicated that emotion-focused coping and seeking social support mediate the relationship between various situational factors and posttraumatic growth (Bellur et al. 2018; He et al. 2013). However, very little is known in the specialized literature about the mediating role of positive coping strategies between traumatic situational factors and PWB in the specific professional context of peacetime military gendarmes. Based on Lazarus and Folkman's (1984) transactional model of stress, this research aims to clarify the role of active, positive coping strategies. In the sense of clearly establishing the psychological or behavioral efforts in the use of own resources that military gendarmes make to face problematic operational and organizational situations, we hypothesized the following:

**Hypothesis 3.1.** *Baseline seeking social support, positive refocusing, and self-control would act as mediators of the relationship between organizational or operational stress and PWB.*

**Hypothesis 3.2.** *Four months apart, seeking social support, positive refocusing, and self-control would act as mediators of the relationship between organizational or operational stress and PWB.*

In order to emphasize the mediating role of perceived social support, seeking social support, positive reappraisal, and self-control, in the relationship between perceived stress and PWB, a longitudinal measurement of them was carried out. The testing was done by analyzing four mediation models. The first model analyzed how OrgS impacted PWB, in a relationship mediated by perceived social support, seeking social support, positive reappraisal, and self-control at T1. The mediating power of the same variables measured at T2 in the relationship between OrgS and PWB was tested in the second model. The third model verified the influence of OpS on PWB in a relationship mediated by perceived social support, seeking social support, positive reappraisal, and self-control, measured at T1. The verification of the mediating role in the relationship between OpS and PWB was achieved by measuring the mediators at T2, in the fourth model. To build these models, perceived stress was measured at T1, PWB four months away (T2), and the mediators, perceived social support, seeking social support, positive reappraisal, and self-control both at T1 and T2. Consecutively, we synthetically present the results of the first two models, and of the next two models in the results section.

## 2. Materials and Methods

### 2.1. Participants and Procedure

The recruitment of participants took place in a meeting attended by all the military personnel of the unit, coordinated by the psychological officer. Following the presentation of the objectives of the study and the two-time testing modality, both male and female military gendarmes expressed their intention to participate. Research was conducted on 210 military personnel, gendarmes, officers, and sub-officers in a military unit, with the following characteristics: all were members of operational staff, including 202 men (96.19%) and 8 women (3.90%), with at least 6 months of experience in the organization. The mean age was 38.52 years (SD = 8.92). The mean duration of military service was 14.52 years (SD = 7.94). The participants are graduates of postgraduate studies (The School of Gendarme Sub-Officers), as well as superior studies (The Police Academy, The Gendarme Faculty). Participants take part in specific peacetime anti-terrorist actions, as well as actions to maintain public order and security. Measures were administered to both executives and leaders. No gendarme has been part of abroad missions for the last two years. From the initial set of data, collected at the time of T1, 11 subjects were eliminated: 3 ended their service reports, 2 gendarmes moved to other units, 3 were not present for the retest, and for 3, the completion of the questionnaires was not compliant.

Based on the informed consent, the participants signed an acceptance agreement before individually completing the questionnaire through the pencil-paper method in an

appropriate environment, with no time limit. For each psychological trial, instructions were given, both verbally and written. The training was identical for each subject and took place at the beginning; the subjects were asked questions regarding the instructions to prove their understanding of the task. The subjects were informed of the purpose of the research and guaranteed anonymity and confidentiality of the data provided, according to the deontological norms of Romanian law.

The questionnaires were translated from English into Romanian by the authors. They included five measures and demographic data (e.g., age, gender, seniority, structure in which they work, type of activity, and mission risk level). The measures were administered in December 2021 for time point T1 and in April 2022 for time point T2. At T1 and T2, instruments were applied to measure PWB, organizational stress, operational stress, coping strategies, and social support, respectively, PWBS, PSQ-Org, PSQ-OP, WCS, and ISEL. The completion time was approximately 30 min. Participation was voluntary, and participants could step away and withdraw from the study at any given moment with no prior explanation needed.

To avoid the priming effect bias that can occur between different subjective measures in the same study (Voicu 2015), we used an alternate sequencing of measures. Specifically, half of the participants were randomly assigned to answer the PWB before the measures of perceived stress (OrgS and OpS), one quarter completing the OrgS first and then the OpS, and the other quarter the other way around. The other half proceeded in the reverse order (first the perceived stress measures and then the PWB). The mediators, social support, and coping mechanisms were assessed at the end of the questionnaire in alternate sequences.

*2.2. Measures*

In the absence of assessment tools specific to the occupational stress of peacetime military gendarmes, this research uses tools developed for police officers.

Organizational stress was measured using the *Organizational Police Stress Questionnaire* (*PSQ-Org*; McCreary and Thompson 2006). The PSQ-Op questionnaire is composed of 20 items, and it evaluates the stress factors associated with the workplace and workplace organization. Each item (e.g., "*Feeling like you always have to prove yourself to the organization*") is evaluated on a scale with 7 points, from 1 ("no stress at all") to 7 ("a lot of stress"), with 4 indicating an average level of stress. The total score was calculated by summing the items, with a higher value indicating a higher level of organizational stress. Participants reported the extent to which they had been affected by organizational stressors in the past 6 months. McCreary and Thompson (2006) reported satisfactory construct validity and reliability for the PSQ-Org. More support for the validity and reliability of this measure has been found in different European samples (e.g., Queirós et al. 2020; Kukić et al. 2021). The Cronbach's alpha value for the present study was 0.937 for time point T1 and 0.950 for time point T2.

Operational stress was measured using the *Operational Police Stress Questionnaire* (*PSQ-OP*, McCreary and Thompson 2006). The PSQ-Op questionnaire, composed of 20 items, evaluates the stress factors associated with specific activities and missions. Each item ("*The risk of being injured while working*") is evaluated on a scale with 7 points, from 1 ("no stress at all") to 7 ("a lot of stress"), with 4 indicating an average level of stress. The total score was calculated by summing the items, with a higher value indicating a higher level of operational stress. The tool measured the intensity of the operational stress factors that the military gendarmes faced in the last 6 months. McCreary and Thompson (2006) have reported satisfactory construct validity and reliability for the PSQ-Op. In a similar European sample, Kukić et al. (2021) found that the reliability of the PSQ-Op was very high, with Cronbach's $\alpha$ being 0.961. The Cronbach's alpha value for the present study was 0.918 for time point T1 and 0.963 for time point T2.

Coping mechanisms were measured using *The Ways of Coping Scale* (*WCS*, Folkman and Lazarus 1988). The WCS questionnaire has been widely used to investigate coping processes in specific, varied contexts and across different populations (Lundqvist and Ahlström 2006; Rexrode et al. 2008). The WCS subscales were designed to provide information regarding

the strengths or weaknesses of coping mechanisms, with major implications for clinical interventions and research (Van Liew et al. 2016). The instrument is composed of 66 items, structured in the following eight dimensions: confrontive coping, distancing, self-control, seeking social support, accepting responsibility, escape-avoidance, planful problem solving, and positive reappraisal. The items are scored on a Likert scale in 4 steps from 0 (not used) to 3 (used a great deal). There were no instructions for participants to use a specific time frame or to focus on a specific event. Participants responded about their overall coping style. Samples of items are "*I tried to keep my feelings to myself*" and "*Accepted sympathy and understanding from someone*". Lundqvist and Ahlström (2006) reported a good internal consistency of the WCQ scales. In this study, only three adaptative coping mechanisms subscales were used: seeking social support, positive reappraisal, and self-control. Total scores were calculated by summing the items corresponding to each subscale. Higher scores indicated higher levels of the analyzed coping strategies. The Cronbach's alpha values for the present study were 0.620 at time T1 and 0.599 at time T2 for the self-control scale, 0.686 at time T1 and 0.724 at time T2 for the seeking social support scale, and 0.611 at time T1 and 0.510 at time T2 for the positive reappraisal scale.

The Psychological Well-Being Scale (*PWBS*, Ryff 1989) was used to measure well-being in military personnel. Participants reported how strongly they agreed or disagreed with 44 statements using a 6-point scale (1 = strongly agree; 6 = strongly disagree). High scores on the PWBS indicate a high degree of PWB. Samples of items are as follows: *The past had ups and downs, but I generally would not want to change it, I enjoy making plans for the future and working to make them a reality, I think it is important to have new experiences that challenge how you think about yourself and the world*. In a similar national sample, researchers did not empirically provide total support for the six-factor model of PWB (Kállay and Rus 2014). Due to the high overlap between the dimensions of PWB, in the present study, we used the PWBS total score, with Cronbach's alpha values of 0.792 for time point T1 and 0.755 for time point T2.

Social support was measured using the *Interpersonal Support Evaluation List* (*ISEL*, Cohen and Wills 1985). This instrument measures the functional components of social support and is composed of 40 items. Each of the following subscales contains 10 items: tangible support, appraisal support, belonging support, and self-esteem support. The respondents indicate the extent to which the items describe the availability of different social support types through a Likert scale in 4 steps (0—definitely false, 1—probably false, 2—probably true, 3—definitely true). No indication was given of specific time periods, reference intervals, or special events. In the present study, general social support was used (the total score). Samples of items include: *If I were sick, I could easily find someone to help me with my daily chores. There is someone I can turn to for advice about handling problems with my family. If I decide one afternoon that I would like to go to a movie that evening, I could easily find someone to go with me*. The ISEL total score was calculated by summing all the items, with a higher value indicating a higher level of social support. The authors reported good construct validity and reliability of the scale (Cohen and Wills 1985). In the present study, we used the ISEL total score, and the Cronbach's alpha values were 0.925 at T1 and 0.955 at T2.

*Sociodemographic variables*. Each participant filled out the solicited information about their age, gender, experience in the organization, and risk degree of missions.

### 2.3. Statistical Analysis

First, descriptive and correlational statistical analyses were created using the software SPSS®, version 28 (IBM Corporation, Armonk, NY, USA). The average values and standard deviations were calculated, and all variables were assessed for normal distributions using Shapiro–Wilk tests. Normality tests showed that all variables were normally distributed ($p > 0.05$). Then, the Pearson correlation coefficients were calculated for the variables of the study.

Second, four longitudinal parallel mediation models were proposed and analyzed using Model 4 (Hayes 2018) from Process version 4.0 with IBM SPSS 28. A total of 5000 bootstrap samples were used, utilizing confidence intervals based on bootstrapping to estimate confidence intervals of 95% (Hayes 2017). Confidence intervals that do not include zero indicate significant effects (Hayes and Scharkow 2013). Mediation analysis was conducted to verify the mediating role of social support and coping mechanisms (seeking social support, positive reappraisal, and self-control), measured at baseline and after four months, in the relationship between baseline OrgS stress or OpS and PWB four months apart.

## 3. Results

### 3.1. Descriptive Statistics and Correlational Analyses

The descriptive statistics and the correlational analyses are presented in Table 1. Both types of stress at baseline, OrgS and OpS, are negatively associated at low and moderate levels with PWB, social support, and coping mechanisms. The participants' age, gender, mission risk, and experience in the organization were not significantly correlated with PWB ($r$s < 0.05, all $p$s > 0.05).

Regarding perceived stress, the results show a significant longitudinal difference in the level of operational stress (MT11 = 34.33; SDT1 = 14.48; MT2 = 36.17; SDT2 = 18.38) compared to organizational stress (MT1 = 33.84; SDT1 = 14.37; MT2 = 33.78; SDT2 = 15.05). Additionally, at an interval of four months, a significant decrease in PWB was found (MT1 = 219.63; SDT1 = 11.84; MT2 = 105.50; SDT2 = 60.71).

### 3.2. Parallel Mediation Analyses of the Relationship between OrgS and PWB

The first model of the study tested both the negative association between baseline OrgS and PWB four months apart and the role of social support and coping mechanisms (social support, positive reappraisal, and self-control) measured at baseline (Table 2) as mediators. Similarly, the second model analyzed the same association, with the same mediators measured after four months (Table 2) (see also Figure 1). The multiple regression analysis was performed to estimate the components of the mediation models.

The results showed that baseline OrgS was negatively associated with social support at T1 ($\beta = -0.0729$; $p < 0.005$) and T2 ($\beta = -0.0915$; $p < 0.001$), seeking social support at T1 ($\beta = -0.0503$; $p < 0.001$) and T2 ($\beta = -0.0401$; $p < 0.005$), positive reappraisal at T1 ($\beta = -0.0520$; $p < 0.007$) and T2 ($\beta = -0.0521$; $p < 0.001$), and self-control at T1 ($\beta = -0.0508$; $p < 0.005$) and T2 ($\beta = -0.0492$; $p < 0.005$). The baseline OrgS was negatively associated with PWB ($\beta = -0.7243$; $p < 0.05$), a relationship partially mediated by social support ($\beta = 3.3216$; $p < 0.001$) measured at T1. Additionally, baseline OrgS was negatively associated with PWB ($\beta = -0.6833$; $p < 0.05$), a relationship partially mediated by social support ($\beta = 2.7342$; $p < 0.001$) and self-control ($\beta = 2.9775$; $p < 0.05$) measured at T2.

Both the direct effect of OrgS (c = $-0.7243$; 95% CI [$-1.2753$; $-0.1734$]) on PWB and the indirect effect of social support (a1 × b1 = $-0.2421$; 95% CI [$-0.4589$; $-0.00359$]) measured at moment T1 are statistically significant since the confidence interval does not include 0. Baseline social support partially mediates the relationship between OrgS and PWB, as the direct effect remains significant. Additionally, both the direct effect of OrgS (f = $-0.6833$; 95% CI [$-1.2345$; $-0.1322$]) and the indirect effect of social support (d1 × e1) = $-0.2502$; 95% CI [$-0.4710$; $-0.00570$]) and of self-control (d4 × e4) = $-0.1464$; 95% CI [$-0.3346$; $-0.0191$]) measured at moment T2 are statistically significant. Social support and self-control partially mediate the relationship between OrgS and PWB, and the direct effect remains significant.

**Table 1.** Means, standard deviations, and correlations between the variables of the study.

| | | Mean | SD | 1 | 2 | 3 | 4 | 5 | 6 | 7 | 8 | 9 | 10 | 11 | 12 | 13 | 14 |
|---|---|---|---|---|---|---|---|---|---|---|---|---|---|---|---|---|---|
| 1 | PWB T1 | 219.63 | 11.82 | 1 | | | | | | | | | | | | | |
| 2 | OrgS T1 | 33.84 | 14.37 | −0.27 ** | 1 | | | | | | | | | | | | |
| 3 | OpS T1 | 34.33 | 14.48 | −0.35 ** | 0.70 ** | 1 | | | | | | | | | | | |
| 4 | PSS T1 | 28.41 | 5.25 | 0.32 ** | −0.19 ** | −0.22 ** | 1 | | | | | | | | | | |
| 5 | PR T1 | 15.65 | 4.00 | 0.25 ** | −0.18 ** | −0.23 ** | −0.04 | 1 | | | | | | | | | |
| 6 | SC T1 | 15.80 | 4.00 | 0.27 ** | −0.18 ** | −0.28 ** | 0.08 | 0.47 ** | 1 | | | | | | | | |
| 7 | SSS T1 | 13.86 | 3.00 | 0.16 * | −0.24 ** | −0.26 ** | 0.24 ** | 0.27 ** | 0.24 ** | 1 | | | | | | | |
| 8 | PWB T2 | 105.50 | 60.71 | 0.86 ** | −0.27 ** | −0.35 ** | 0.34 ** | 0.19 ** | 0.23 ** | 0.19 ** | 1 | | | | | | |
| 9 | OrgS T2 | 33.78 | 15.05 | −0.27 ** | 0.94 ** | 0.66 ** | −0.22 ** | −0.20 ** | −0.20 ** | −0.25 ** | −0.32 ** | 1 | | | | | |
| 10 | OpS T2 | 36.17 | 18.38 | −0.32 ** | 0.68 ** | 0.91 ** | −0.25 ** | −0.22 ** | −0.22 ** | −0.25 ** | −0.36 ** | 0.68 ** | 1 | | | | |
| 11 | PSS T2 | 29.20 | 6.15 | 0.25 ** | −0.21 ** | −0.19 ** | 0.89 ** | 0.01 | 0.06 | 0.24 ** | 0.34 ** | −0.27 ** | −0.25 ** | 1 | | | |
| 12 | PR T2 | 16.58 | 3.21 | 0.24 ** | −0.23 ** | −0.25 ** | 0.10 | 0.92 ** | 0.45 ** | 0.33 ** | 0.21 ** | −0.28 ** | −0.29 ** | 0.10 | 1 | | |
| 13 | SC T2 | 16.42 | 3.51 | 0.26 ** | −0.20 ** | −0.34 ** | 0.12 | 0.44 ** | 0.93 ** | 0.27 ** | 0.27 ** | −0.22 ** | −0.30 ** | 0.12 | 0.45 ** | 1 | |
| 14 | SSS T2 | 14.46 | 2.80 | 0.12 | −0.20 ** | −0.19 ** | 0.23 ** | 0.20 ** | 0.17 ** | 0.92 ** | 0.17 * | −0.24 ** | −0.23 * | 0.27 ** | 0.26 ** | 0.21 ** | 1 |

Note: * $p < 0.05$; ** $p < 0.01$; PWB—psychological well-being; OrgS—organizational stress; OpS—operational, PSS—perceived social support, PR—positive reappraisal, SC—self-control, SSS—seeking social support.

**Table 2.** Direct, indirect, and total effects of the relationship between OrgS at T1 and PWB at T2, mediated by perceived social support, seeking social support, positive reappraisal, and self-control, mediators that act at moment T1 and T2.

| | *B* | **SE** | *p* | **95% CI** |
|---|---|---|---|---|
| **Total effects** | | | | |
| OrgST1—PWB (mediators T1) | −0.724 | 0.279 | 0.010 | |
| OrgST1—PWB (mediators T2) | −0.683 | 0.279 | 0.015 | |
| **Direct effects** | | | | |
| OrgST1—Perceived social support T1 ($M_1$) | −0.072 | 0.024 | 0.003 | |
| OrgST1—Seeking social support T1 ($M_2$) | −0.050 | 0.014 | <0.001 | |
| OrgST1—Positive reappraisal T1 ($M_3$) | −0.052 | 0.019 | 0.006 | |
| OrgST1—Self—control T1 ($M_4$) | −0.050 | 0.019 | 0.008 | |
| OrgST1—Perceived social support T2 ($M_5$) | −0.091 | 0.029 | 0.001 | |
| OrgST1—Seeking social support T2 ($M_6$) | −0.040 | 0.013 | 0.002 | |
| OrgST1—Positive reappraisal T2 ($M_7$) | −0.052 | 0.015 | <0.001 | |
| OrgST1—Self—control T2 ($M_8$) | −0.049 | 0.016 | 0.003 | |
| **Indirect effects** | | | | |
| OrgST1—Perceived social support T1 ($M_1$)—PWB | −0.242 | 0.105 | <0.001 | [−0.458; −0.037] |
| OrgST1—Seeking social support T1 ($M_2$)—PWB | −0.028 | 0.079 | 0.688 | [−0.198; 0.129] |
| OrgST1—Positive reappraisal T1 ($M_3$)—PWB | −0.057 | 0.067 | 0.320 | [−0.214; 0.059] |
| OrgST1—Self—control T1 ($M_4$)—PWB | −0.105 | 0.077 | 0.058 | [−0.294; 0.007] |
| OrgST1—Perceived social support T2 ($M_5$)—PWB | −0.250 | 0.106 | <0.001 | [−0.475; −0.056] |
| OrgST1—Seeking social support T2 ($M_6$)—PWB | −0.008 | 0.065 | 0.884 | [−0.139; 0.133] |
| OrgST1—Positive reappraisal T2 ($M_7$)—PWB | −0.068 | 0.077 | 0.335 | [−0.234; 0.074] |
| OrgST1—Self—control T2 ($M_8$)—PWB | −0.146 | 0.080 | 0.016 | [−0.326; −0.155] |

Note: PWB—psychological well-being; OrgS—organizational stress; $M_1$—perceived social support at T1; $M_2$—seeking social support at T1; $M_3$—positive reappraisal at T1; $M_4$—self—control at T1; $M_5$—perceived social support at T2; $M_6$—seeking social support at T2; $M_7$—positive reappraisal at T2; $M_8$—self—control at T2.

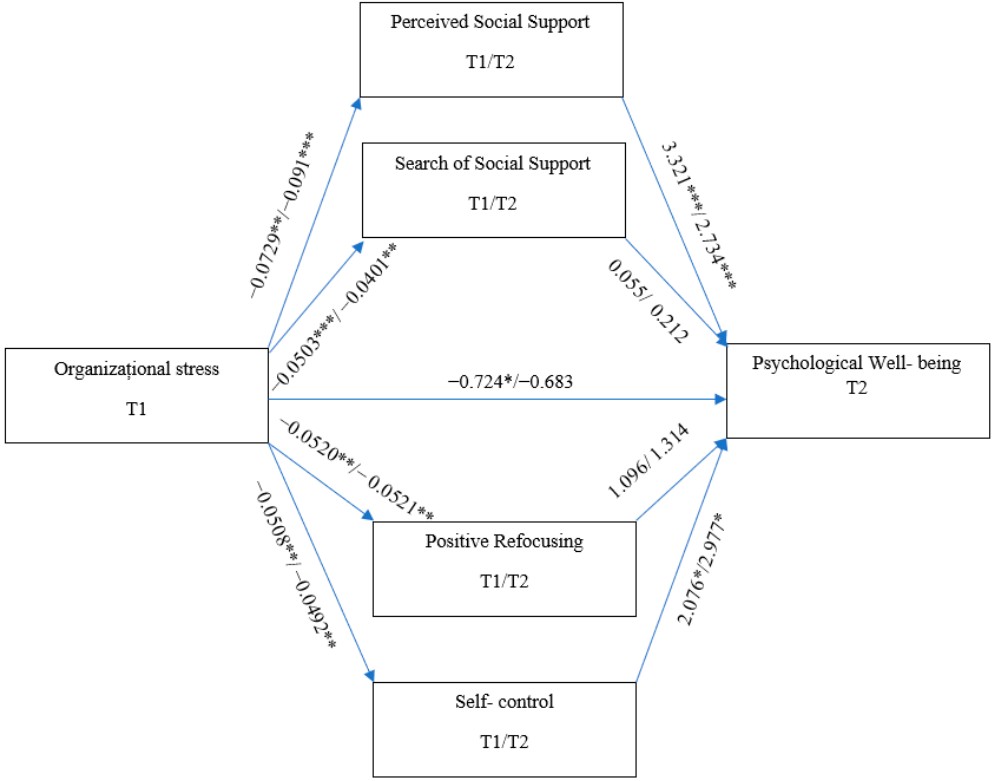

**Figure 1.** The relationship between organizational stress and psychological well-being with mediators at T1 and T2. * $p < 0.05$. ** $p < 0.01$. *** $p < 0.001$.

### 3.3. Parallel Mediation Analyses of the Relationship between OpS and PWB

The third model of the study tested both the negative association between baseline OpS stress and PWB four months apart and the role of social support and coping mechanisms (seeking social support, positive reappraisal, and self-control) as mediators, measured at baseline (Table 3). Similarly, the fourth model analyzed the same association and the role of the same mediators, measured four months apart (Table 3) (see also Figure 2). The analysis of the multiple regression was undertaken to estimate the components of the mediation model.

**Table 3.** Direct, indirect, and total effects of the relationship between OpS at T1 and PWB at T2, mediated by perceived social support, seeking social support, positive reappraisal, and self-control, mediators that act at moment T1 and T2.

| | *B* | SE | *p* | 95% CI |
|---|---|---|---|---|
| Total effects | | | | |
| OpST1—PWB (mediators T1) | −1.034 | 0.281 | <0.001 | |
| OpST1—PWB (mediators T2) | −1.018 | 0.281 | <0.001 | |
| Direct effects | | | | |
| OpST1—Perceived social support T1 ($M_1$) | −0.082 | 0.024 | <0.001 | |
| OpST1—Seeking social support T1 ($M_2$) | −0.053 | 0.013 | <0.001 | |
| OpST1—Positive reappraisal T1 ($M_3$) | −0.063 | 0.018 | <0.001 | |
| OpST1—Self—control T1 ($M_4$) | −0.079 | 0.018 | <0.001 | |
| OpST1—Perceived social support T2 ($M_5$) | −0.084 | 0.028 | 0.003 | |
| OpST1—Seeking social support T2 ($M_6$) | −0.038 | 0.013 | 0.003 | |
| OpST1—Positive reappraisal T2 ($M_7$) | −0.057 | 0.014 | <0.001 | |
| OpST1—Self—control T2 ($M_8$) | −0.084 | 0.015 | <0.001 | |
| Indirect effects | | | | |
| OpST1—Perceived social support T1 ($M_1$)—PWB | −0.258 | 0.104 | <0.001 | [−0.474; −0.067] |
| OpST1—Seeking social support T1 ($M_2$)—PWB | −0.020 | 0.085 | 0.782 | [−0.192; 0.158] |
| OpST1—Positive reappraisal T1 ($M_3$)—PWB | −0.064 | 0.077 | 0.354 | [−0.236; 0.074] |
| OpST1—Self—control T1 ($M_4$)—PWB | −0.125 | 0.095 | 0.150 | [−0.333; 0.050] |
| OpST1—Perceived social support T2 ($M_5$)—PWB | −0.225 | 0.108 | <0.001 | [−0.448; −0.021] |
| OpST1—Seeking social support T2 ($M_6$)—PWB | −0.006 | 0.064 | 0.909 | [−0.132; 0.139] |
| OpST1—Positive reappraisal T2 ($M_7$)—PWB | −0.074 | 0.081 | 0.330 | [−0.244; 0.083] |
| OpST1—Self—control T2 ($M_8$)—PWB | −0.177 | 0.108 | 0.091 | [−0.396; 0.034] |

Note: PWB—psychological well-being; OpS—operational stress; $M_1$—perceived social support at T1; $M_2$—seeking social support at T1; $M_3$—positive reappraisal at T1; $M_4$—self—control at T1; $M_5$—perceived social support at T2; $M_6$—seeking social support at T2; $M_7$—positive reappraisal at T2; $M_8$—self—control at T2.

The results showed that baseline OpS was negatively associated with social support at T1 ($\beta = -0.0826$; $p < 0.001$) and T2 ($\beta = -0.0845$; $p < 0.005$), seeking social support at T1 ($\beta = -0.0539$; $p < 0.001$) and T2 ($\beta = -0.0385$; $p < 0.005$), positive reappraisal at T1 ($\beta = -0.0637$; $p < 0.001$) and T2 ($\beta = -0.0572$; $p < 0.001$) and self-control at T1 ($\beta = -0.0796$; $p < 0.001$) and T2 ($\beta = -0.0845$; $p < 0.001$). Additionally, baseline OpS was negatively associated with PWB ($\beta = -1.0344$; $p < 0.001$), a relationship mediated by social support ($\beta = 3.1285$; $p < 0.001$) measured at moment T1. In addition, baseline OpS was negatively associated with PWB ($\beta = -1.0187$; $p < 0.001$), and this relationship was mediated by social support ($\beta = 2.6656$; $p < 0.001$) measured at moment T2.

Both the direct effect of OpS (c′ = −1.0344; 95% CI [−1.5889; −0.4798]) and the indirect effect of social support measured at moment T1 (a1′ × b1′ = −0.2584; 95% CI [−0.4712; −0.0637]) are statistically significant, as the confidence interval does not include 0. Thus, social support (T1) partially mediates the relationship between OpS and PWB. Additionally, both the direct effect of OpS (f′ = −1.0187; 95% CI [−1.5738; −0.4637]) on PWB and the indirect effect of social support (d1′ × e1′) = −0.2252; 95% CI [−0.4519; −0.00234]) measured at moment T2 are statistically significant. This proves that social support partially mediates the relationship between OpS and PWB.

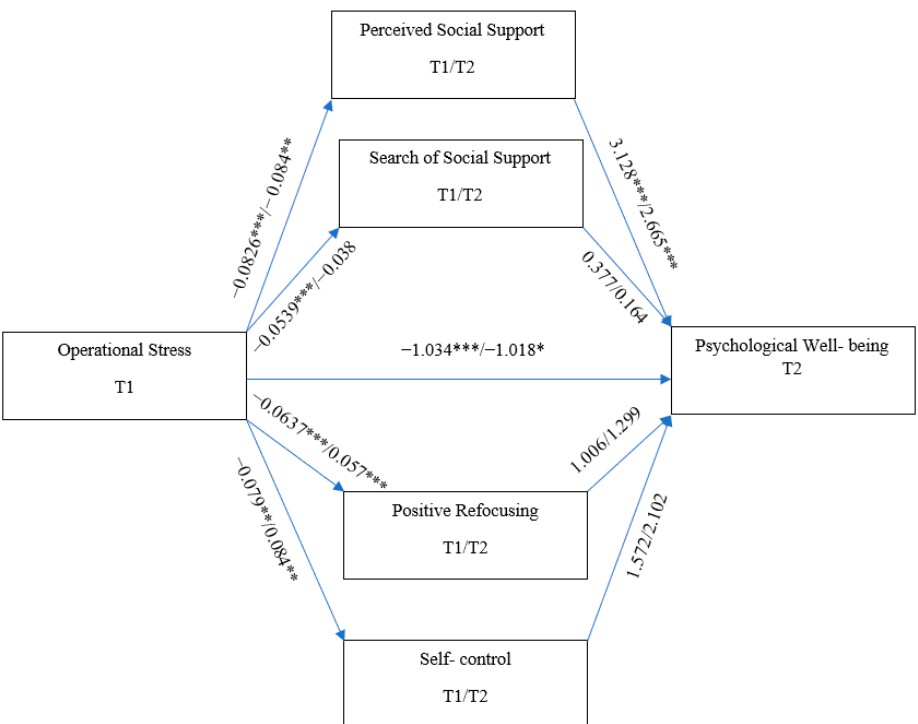

**Figure 2.** The relationship between operational stress and psychological well-being with mediators at T1 and T2. * $p < 0.05$. ** $p < 0.01$. *** $p < 0.001$.

## 4. Discussion

### 4.1. Theoretical Implications

The present study aimed to investigate the longitudinal association of perceived work-related stress with the PWB of military gendarmerie personnel in peacetime activities from a southeastern European country. Moreover, the study investigated the mediating role of social support and coping mechanisms in the relationship between OrgS and OpS with PWB. Four parallel mediation models were tested to investigate the mechanisms explaining the relationship between perceived work-related stress and the PWB of military personnel. Thus, we investigated some of the personal and organizational contexts that can affect psychological well-being in gendarmes.

Preliminary analysis indicated that both types of stress, OrgS and OpS, are negatively associated with PWB, social support, positive refocusing, self-control, and search for social support. OrgS shows low and negative correlation with PWB, while OpS is moderately and negatively correlated with PWB. These findings contradict some previous research indicating that OrgS may be more strongly associated with well-being than OpS in low enforcement forces because their work environment is perceived as oppressive (Shane 2010) and under-rewarded (Basinska and Wiciak 2013). However, OrgS factors could be more easily underestimated in participants' reports compared to OpS factors.

*The relationship between OrgS and PWB.* As hypothesized, the results showed a negative direct correlation between OrgS at baseline and PWB measured four months apart. The present study confirms the fact that military personnel of the gendarmerie could be considered a group with a high risk of developing psychological disorders because of the many stress factors they face in the workplace. The results are consistent with previous research indicating that different aspects of public order and security activity could significantly affect the PWB of military and police personnel. For example, Hart et al. (1995) mentioned that OrgS was the most important predictor of well-being and decreased quality of life in police personnel. OrgS factors have been identified as being likely to cause psychological turmoil among personnel who work in the field of public order and security (Kop and Euwema 2001; Tyagi and Dhar 2014). The schedule, the large amount of work, the culture,

and organizational changes have a significant impact on the mental state of the mentioned personnel (Purba and Demou 2019). The high demands of this operational group are associated with burnout, utilizing force more often, poor interaction with the public, health issues, tense relationships, and a low quality of work (Martinussen et al. 2007). Research mentions that OrgS factors have such a significant effect on PWB due to the rigidity and bureaucratic nature of the organization, as well as the resistance to change and the refusal to correct traditional practices (Stinchcomb 2004).

*The relationship between OpS and PWB.* The present study hypothesized and confirmed that baseline OpS is directly and negatively correlated with PWB measured four months apart. OpS has its origin in high-risk missions, unpredictability of duties, activities of great importance, sometimes threatening, confrontations with public hostility or insults, direct threats of death, or threats regarding family and personal integrity. This result is consistent with previous research indicating that OpS factors represent a main source of stress (Garcia et al. 2004; Lucas et al. 2012; Violanti et al. 2016). These studies mention that higher stress levels associated with the risk of injury from trauma, exposure to physical assault, and the risk of being murdered are negatively associated with military personnel well-being.

The present study demonstrates that perceived work-related stress, OrgS and OpS, are significant negative predictors of PWB, extending previous research on military gendarmes in their peacetime activities. These results confirm the first hypothesis and support the results of previous research suggesting that work in the field of public order and security is one of the most stressful, with a high impact on the physical and mental well-being of employees, life and job satisfaction, efficacy, and self-perception (Garbarino et al. 2013; Johnson et al. 2005).

*Social support acting as a mediator.* The relationships between the study variables were partially mediated by social support as an explanatory mechanism. Social support measured at baseline and four months apart are significant mediators of the longitudinal relationship between both OrgS and PWB, and OpS and PWB. Therefore, military gendarmes who are more stressed at work report not only lower levels of PWB four months apart, but also lower baseline and four-month levels of perceived social support. This is consistent with one of the main assertions of the damaged social support model (Dean and Ensel 1982; Lin and Dean 1984), indicating that stress deteriorates the perception of available social relationships, which diminishes PWB. Perceived available social support is important in the long term, producing significant indirect effects on the relationship between Org/OpS and PWB. Our findings indicate that the effect of work stress on perceived social support is maintained longitudinally, continuing to have an effect on PWB. At the same time, social support acts in a direct, positive, continuous way on PWB, becoming one of the most important external resources in facing work stress. Social support received from family, friends, coworkers, and bosses constitutes a continuous positive predictor of PWB in military gendarmes. The results underline the importance of perceived social support in enhancing PWB by influencing feelings, cognitions, and positive behaviors (Cohen et al. 2000). Social support can thus provide the feeling of belonging, safety, and identity to military gendarmes, especially when facing organizational stress factors, therefore causing a higher level of PWB through beneficial interaction. These results underline the importance of perceived support from friends, family, or coworkers, which can contribute to decreasing the impact of stress on PWB. They are consistent with previous research suggesting that social support is associated with many positive results regarding mental and physical health (Holt-Lunstad et al. 2010; Uchino et al. 2012). For police officers, for example, social support has been associated with a reduced perception of work-related stress (Vig et al. 2020), improvement of general mental health (Hansson et al. 2017), and an increase in workplace satisfaction level (Brough and Frame 2004). Officers and sub-officers are extremely vulnerable if they lack social support (family, friends, coworkers) or the personal ability to deal with intense demands (Loriol 2016). Without adequate support, police and military personnel can feel powerless or even marginalized, stigmatized, and isolated (Bell and Eski 2016). Our study demonstrates that the military organization of the Romanian

Gendarmerie continues to be guided by the parameters of traditional values that define team spirit, including social support received and provided. Gendarmes manage to meet the challenges of the end of the millennium and cultivate their military cultural values.

*Coping mechanisms as mediators.* Concerning coping mechanisms, self-control measured after four months acts as a mediator only in the relationship between OrgS and PWB. OrgS has a significantly negative longitudinal effect on self-control, and after four months, self-control positively correlates with PWB. Military gendarmes with a higher level of OrgS will show less self-control in comparison to those with a lower level, and vice versa. Although some research has mentioned a positive association between self-control and stress (Schilling et al. 2022), the present study indicates their negative correlations and the role of self-control after four months in the links between OrgS and PWB. The mediation effect of self-control on PWB is partial, along with that of social support. This result is consistent with previous research suggesting that self-control is a coping mechanism used frequently by police officers when facing aversive events and trauma (Violanti 1993). Garbarino et al. (2013) mention, for example, that police officers must show courage as well as physical and emotional strength since the loss of emotional control is considered a weakness (Garbarino et al. 2013), which could lead to the suppression of feelings (Bell and Eski 2016). Military gendarmes with high levels of self-control are more able to exercise their reflective processes and pursue their long-term goals, which are strong predictors of PWB (Luciano et al. 2004). Better acceptance of emotions and stress by gendarmes makes them less likely to use avoidant coping strategies that affect their mental health, leading to the development of PTSD symptoms (Carleton et al. 2022). Insignificant results were obtained regarding the mediating role of positive refocusing and seeking social support.

### 4.2. Theoretical and Practical Implications

As the model of well-being in military predicts, work demand (such as job stress) influences well-being, and this link is explained through different personal, organizational, and community resources (Bowles et al. 2017). Our study expands current knowledge by indicating that, despite the fact that OrgS is more weakly associated with PWB, compared to OpS, several psychological mechanisms explain its link with PWB, not only the baseline and after four months apart perceived SS, but also the seeking SS measured after four months, as a coping mechanism.

The results may have important practical implications in improving the PWB of military gendarmes by influencing empirical psychosocial interventions. First, the study could provide the basis for strengthening intervention programs aimed at raising awareness of the importance of identifying and mitigating OrgS and OpS factors to reduce their impact on military personnel's PWB. Second, interventions should focus on increasing perceived social support. Educating staff about the types of behaviors and support that are beneficial in the work of the gendarmes could help to increase the perception of support available and the perception of support given and received within the organization, which could have a direct positive effect on PWB. Third, managers and mental health professionals could create an organizational culture based on mutual support through intervention programs. Fourth, an important role that mental health specialists have is to promote and educate on the most effective coping mechanisms that military gendarmes can use when faced with stressors. Moreover, significant consideration must be given to vulnerable military personnel likely to experience reduced well-being. Therefore, developing intervention strategies focused on awareness of the importance of social support, increasing awareness of available support, understanding the value of the provider, and developing the ability to provide support within the military organization are very important. In order to positively influence psychological well-being, attitude, and behavior, this study directs organizations to develop training programs that help gendarmes understand the values and philosophy of the culture, based on team spirit and mutual support. In this sense, to prevent a clash of values (Veinhardt and Gulbovaite 2017), which can decrease employees' identification with the military organization, research supports the need for congruence between the form and

content of training (Ben-Hador et al. 2020), guiding researchers and practitioners towards the cultivation of real values of social support. Moreover, strengthening healthy behavior and adaptive coping strategies or replacing maladaptive coping with adaptive coping are important in training effective military personnel (Cahill et al. 2021).

*4.3. Limitations and Future Directions*

Although the present study offers additional information on the relationships of OrgS and OpS with PWB in military gendarmes, explained through the mediating role of social support and certain coping mechanisms, a series of limitations can be noted. First, the associations between the variables of the present study are based on data generated through quantitative measurements, which offers restrictive information on how and why the analyzed variables lead to an increase in the PWB of military gendarmes. The use of self-report scales hinders the observation of real manifestations of the analyzed processes. Second, the findings are based on measuring perceptions of a moderate number of military employees due to the special numerous approvals that are difficult to obtain to carry out studies on this group of participants. Therefore, the results of the present study must be interpreted in the context of the specific sample used. Larger-scale studies, with a larger range of participants and using multiple waves of measurements, could retest the present study's results. Finally, investigating work-related stress and PWB in the military field is difficult. Military structures are based on a strong organizational culture to which strong personalities adhere. In this context, showing one's emotions could be considered a weakness, making the task of collecting accurate data difficult.

## 5. Conclusions

In conclusion, the present study investigated the longitudinal relationship between OrgS and OpS with PWB after four months, mediated by social support and coping mechanisms within a gendarmes' sample. Both OrgS and OpS reported by military personnel are negatively associated with PWB. The perceived social support reported at both T1 and T2 partially mediates the relationship between OrgS and PWB and the relationship between OpS and PWB. Among coping mechanisms, only the self-control reported after four months (T2) was a significant mediator in the relationship between OrgS and PWB, but not between OpS and PWB. The present study contributes to a better understanding of the explanatory mechanism of the relationship between work-related stress and PWB. The fact that social support measured in both moments and self-control measured four months apart act as explanatory variables for understanding the way military personnel's OrgS and OpS predict PWB has significant practical implications.

**Author Contributions:** Conceptualization, M.N.T. and A.-D.B.; methodology, M.N.T. and A.-D.B.; software, A.-D.B.; validation, A.-D.B.; formal analysis, A.-D.B.; investigation, A.-D.B.; resources, M.N.T. and A.-D.B.; data curation, A.-D.B.; writing—original draft preparation, M.N.T. and A.-D.B.; writing—review and editing, M.N.T.; visualization, A.-D.B.; supervision, M.N.T.; project administration, M.N.T. All authors have read and agreed to the published version of the manuscript.

**Funding:** This research received no external funding.

**Institutional Review Board Statement:** The study was conducted in accordance with the Declaration of Helsinki, and approved by the Ethics Committee of Alexandru Ioan Cuza (protocol code 1031 /12.07.2023).

**Informed Consent Statement:** Informed consent was obtained from all subjects involved in the study.

**Data Availability Statement:** The data that support the findings of this study are available on request from the corresponding author due to privacy issues.

**Conflicts of Interest:** The authors declare no conflict of interest.

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
