# Peer review of "Stress and Psychological Well-Being in Military Gendarmes"

_socsci, doi:10.3390/socsci12090517_

Round 1

Reviewer 1 Report

Although the document presents an exciting theme to analyze, it still needs to be presented what the knowledge gap or the research vacuum these findings contribute to science, which will be developed in the introduction. On the other hand, the discussion of the results could be more robust based on the results and the established hypotheses. It is suggested to broaden the theoretical implications but mainly the practical implications. The conclusions are general and do not respond to the hypotheses identified. Only three references from 2020 and 2 from 2021 were found. I suggest including current information that better justifies the development of the hypotheses.

English is understandable but could be improved with a language edit.

Author Response

Thank you for your important suggestions! We have considered each recommended suggestion.

       “Although the document presents an exciting theme to analyze, it still needs to be presented what the knowledge gap or the research vacuum these findings contribute to science, which will be developed in the introduction. On the other hand, the discussion of the results could be more robust based on the results and the established hypotheses. It is suggested to broaden the theoretical implications but mainly the practical implications. The conclusions are general and do not respond to the hypotheses identified. Only three references from 2020 and 2 from 2021 were found. I suggest including current information that better justifies the development of the hypotheses”.

We want to thank Reviewer #1 for these suggestions. Regarding the first opinion, regarding the research gap we added: Also, the explanatory mechanisms of this relationship are not yet sufficiently investigated. Moreover, most of the previous research investigated the link between work-related stress and well-being cross-sectionally, and less longitudinally. (lines 56-58). We have also added, that the topic is extremely relevant because:The Gendarmerie has a unique status in Romanian society, being both an institution that ensures security and one with a symbolic capital or unique form of cultural militarism. In the context of this new millennium, a civil-military gap begins to manifest in military organizations (Ruckert, 2006), a dissociation from the military ethos in favor of a new individualistic, post-modern ethos. The military is beginning to become more and more similar to civilian organizations (Lebel et al., 2022). In the context of these transformations, the study analyzes the perspective of stress in relation with the well-being, perceived organizational social support and the coping strategies, perceived by military gendarmes as being rewarding, and which defines supportive relationship and fighting spirit.” (lines 61-70).

“Psychological well-being is extremely important for the success of military operations, for the mental and physical health of military personnel (Chen at all, 2018).” (lines 83-85)

“Social support has pervasive benefits throughout adulthood (Shin & Park, 2022). However, its importance could be lost over time due to changes in age or social circumstances associated (Carstensen, 2006) with the military profession. It is difficult to assess the importance of different psychological needs for psychological well-being (Vermote et al., 2022), and at the level of the military gendarmerie population this information is missing (lines 158-163).

Regrading the extension of theoretical implications, we have added:

“As the model of well-being in military predict, work-demand (such as job stress) influence well-being, and this link is explained through different personal, organizational and community resources (Bowels at al., 2017). Our study expands current knowledge by indicating that, despite the fact that OrgS is more weakly associated with PWB, compared to OpS, more psychological mechanisms explain its connection with PWB, not only the baseline and after four moth apart perceived SS, but also the seeking SS after four months, as coping mechanism.” (lines 623-640)

Reviewer 2 Report

This is a classic article in the field of organizational behavior, which attempts to link variables that can be defined as:

A. internal (institutional)

B. Measurers (for which there is an agreement on the methods of measuring them using agreed-upon questionnaires in the literature)

C. rooted in organizational psychology (although some are objective, and some are subjective).

The article is original in that it teaches shattering some axioms, for example about the function attributed to Social Support.

The article is solidly written and is a conservative report on the study of organizational behavior.

I can certainly recommend its publication, but I find its summary narrow. The authors need to include some references to several additional research options that are worth mentioning. Those that they did not take but deserve to be mentioned in terms of follow-up studies.

To illustrate to the readers the research potential that exists in the empirical data repertoire that the article holds.

For example, the study of Psychological Well Being are much being studied nowadays in the perspective of sense-making, sense of commitment, the discourse of meaning, and other components of a positive psycho among military personnel. All those practices related to what is known as "Discursive Practices" are measured by "Institutional Discursive Analysis". The intention is to examine the context that these military personnel feel in relation to what is known as the "Calling theory" which acts to empower them through a "Calling Discourse", that provides meaning reward, and other forms of capital.

For reading on the subject, which can also be used as a source for a quote in the article, see:

Lebel, U., Ben-Hador, B. and Ben-Shalom, U., "Jewish Spiritualization as a 'Meaning Injection': an Ethnography of Rabbinic Seminary in the Israeli Military", in: Mayseless, O. and Russo-Netzer , P. (eds), 2022, Finding Meaning: An Existential Quest in Postmodern Israel, Oxford University Press, NY, pp. 355-375.

And

Ben-Hador, B., Lebel, U. and Ben-Shalom, U. “Learning How to Lead from King David? On the Gap between Declared and Real Content in Training”. European Journal of Training and Development 44(4-5), 2020, 489-507.

Mention of this direction as well as others for examining the variables that affect the Psychological Well Being of military personnel, or to influence on

Psychological Well-Being of them - can turn the article from a research report to an article that, beyond its empirical effort, also provides a contribution to those interested in being exposed to the study of the field from the intellectual and theoretical side.

Author Response

Thank you for the important suggestions! We have considered each recommended suggestion.

„The article is solidly written and is a conservative report on the study of organizational behavior. 

I can certainly recommend its publication, but I find its summary narrow. The authors need to include some references to several additional research options that are worth mentioning. Those that they did not take but deserve to be mentioned in terms of follow-up studies.

To illustrate to the readers the research potential that exists in the empirical data repertoire that the article holds.

For example, the study of Psychological Well Being are much being studied nowadays in the perspective of sense-making, sense of commitment, the discourse of meaning, and other components of a positive psycho among military personnel. All those practices related to what is known as "Discursive Practices" are measured by "Institutional Discursive Analysis". The intention is to examine the context that these military personnel feel in relation to what is known as the "Calling theory" which acts to empower them through a "Calling Discourse", that provides meaning reward, and other forms of capital”.

In relations to your comment that the summary is narrow we added: „Work-related stress is a significant determinant of the psychological well-being, but more relevant are the nature of stressors that military personnel are facing, and the factors than can explain the relationship between work stress and well-being. (lines 8-10).

Regarding the examination of the context, which gives meaning to the military gendarmes, we added: The Gendarmerie has a unique status in Romanian society, being both an institution that ensures security and one with a symbolic capital or unique form of cultural militarism. In the context of this new millennium, a civil-military gap begins to manifest in military organizations (Ruckert, 2006), a dissociation from the military ethos in favor of a new individualistic, post-modern ethos. The military is beginning to become more and more similar to civilian organizations (Lebel et al., 2022). In the context of these transformations, the study analyzes the perspective of stress in relation with the well-being, perceived organizational social support and the coping strategies, perceived by military gendarmes as being rewarding, and which defines supportive relationship and fighting spirit.” (lines 54-63).

In terms of directing researchers and practitioners to examine other variables that may affect psychological well-being, we added: “In order to positively influence psychological well-being, attitude and behavior, this study directs organizations to develop training programs that help gendarmes understand the values ​​and philosophy of the culture, based on team spirit and mutual support. In this sense, to prevent a clash of values ​​(Veinhardt & Gulbovaite, 2017), which can decrease employees' identification with the military organization, research supports the need for congruence between the form and content of training (Ben-Hador et al., 2020), guiding researchers and practitioners towards the cultivation of real values ​​of social support. Moreover, strengthening healthy behavior and adaptive coping strategies or replacing maladaptive coping with adaptive coping are important in training effective military personnel (Cahill et al, 2021).” (lines 648-657).

We have included new references: Bowels at al., 2017, Cahill et al, 2021, Shin & Park, 2022; Vermote et al., 2022; Vig et al., 2020; Lebel et al., 2022.

Round 2

Reviewer 1 Report

Thank you very much for the adjustments made in the document. I have no further observations. Good job!

About the language, just a minor revision. 

Author Response

Thank you for all your usefull suggestions. We have checked the spelling errors once more.